# Structural basis of *S*-adenosylmethionine-dependent allosteric transition from active to inactive states in methylenetetrahydrofolate reductase

Kazuhiro Yamada [1,2,3] ✉, Johnny Mendoza [1] & Markos Koutmos [1,2] ✉

Methylenetetrahydrofolate reductase (MTHFR) is a pivotal flavoprotein connecting the folate and methionine methyl cycles, catalyzing the conversion of methylenetetrahydrofolate to methyltetrahydrofolate. Human MTHFR (*h*MTHFR) undergoes elaborate allosteric regulation involving protein phosphorylation and *S*-adenosylmethionine (AdoMet)-dependent inhibition, though other factors such as subunit orientation and FAD status remain understudied due to the lack of a functional structural model. Here, we report crystal structures of *Chaetomium thermophilum* MTHFR (*c*MTHFR) in both active (R) and inhibited (T) states. We reveal FAD occlusion by Tyr361 in the T-state, which prevents substrate interaction. Remarkably, the inhibited form of *c*MTHFR accommodates two AdoMet molecules per subunit. In addition, we conducted a detailed investigation of the phosphorylation sites in *h*MTHFR, three of which were previously unidentified. Based on the structural framework provided by our *c*MTHFR model, we propose a possible mechanism to explain the allosteric structural transition of MTHFR, including the impact of phosphorylation on AdoMet-dependent inhibition.

Methylenetetrahydrofolate reductase (MTHFR) catalyzes the enzymatic reduction of methylenetetrahydrofolate (CH₂-H₄folate) using NAD(P)H as the reducing agent[1,2]. MTHFR serves as the major source of methyltetrahydrofolate, which is the substrate of the downstream enzyme, methionine synthase[3]. Methionine synthase catalyzes the formal methyl transfer from methyltetrahydrofolate to homocysteine, yielding tetrahydrofolate and methionine as products[4,5]. Methionine is thereafter converted to *S*-adenosylmethionine (AdoMet), which is known as biology's universal methyl donor. In addition, AdoMet acts as an allosteric regulator of MTHFR and a feedback loop regulator in folate and methionine metabolism (Fig. 1a)[6–8]. The regulation of MTHFR activity via AdoMet is crucial in controlling the flux of folate-carrying one-carbon units to methionine synthesis, helping bridge the

folate and methionine cycles, as it is the only known enzyme to exhibit AdoMet-dependent inhibition[9].

MTHFR non-covalently binds FAD as an essential cofactor, shuttling electrons from NAD(P)H in a ping-pong bi-bi mechanism[10]. The eukaryotic form of MTHFR is homodimeric, where each monomer consists of the N-terminal catalytic domain connected by a linker to the C-terminal regulatory domain (Fig. 1b)[7]. In contrast, bacterial MTHFR lacks the regulatory domain and manifests as a homooligomer consisting only of catalytic subunits[11–13]. The catalytic component of most MTHFRs binds FAD within the core of a $\beta_8\alpha_8$ barrel structure, also known as a TIM barrel. The regulatory module has a distinct structural topology that facilitates the formation of MTHFR dimers through the interface of its regulatory domains[7]. This domain has been shown to

[1]Department of Chemistry, University of Michigan, Ann Arbor, MI 48109, USA. [2]Program in Biophysics, University of Michigan, Ann Arbor, MI 48109, USA. [3]Present address: Department of Biological Chemistry, University of Michigan, Ann Arbor, MI 48109, USA. ✉e-mail: yamadak@umich.edu; mkoutmos@umich.edu

**Fig. 1 | Folate and methionine metabolism one-carbon cycle. a** One carbon units (C1-units, such as formyl, methenyl, methylene, or methyl group) carried by folate are supplied from amino acids (Gly, Ser, and His). C1-units and metabolites incorporating these one-carbon units are shown in red. C1-units are used to synthesize purine bases, dTMP, and methionine (Met). Methionine is synthesized to AdoMet, which is the major methyl donor in the cell. AdoMet is an allosteric inhibitor for MTHFR, whose activity determines the flux of folate C1-unit to methionine synthesis and transmethylation. **b** Domain organization of MTHFRs. The catalytic domain of MTHFR contains FAD as an essential cofactor (shown in yellow). Although bacterial MTHFRs only consist of catalytic subunits, eukaryotic MTHFRs consist of both a catalytic domain and a regulatory domain, connected by a linker. AdoMet binds to the regulatory domain. Human (*Homo sapiens*) MTHFR has a Ser/Thr-rich region on its N-terminus which can be phosphorylated (shown in circled P with pink), enhancing its sensitivity for allosteric regulation by AdoMet. In this study, MTHFR from *Chaetomium thermophilum* is used as a structural model for *h*MTHFR, with which it shares reasonable homology. **c** Phosphorylation mapping of *h*MTHFR. Phosphorylation sites were identified by our analysis using tryptic fragments and LS-MS/MS analysis (indicated by circled P in pink). Overlapping phosphorylation sites in both results, which occurred on eight amino acid side chains, are represented by red circled P. Phosphorylation occurs intensively in the N-terminal Ser-Thr rich site and the C-terminal end of the linker.

bind *S*-Adenosyl-L-homocysteine (AdoHcy) and is presumed to bind AdoMet, though no structure of the AdoMet bound form currently exists.

MTHFR has evolved several forms of (allosteric) regulation, ranging from substrate-induced inhibition in bacteria (NADH)[12], NADH preference in bacteria as opposed to NADPH preference in eukaryotes[14–16], to the more complex AdoMet-dependent allosteric inhibition observed in eukaryotes[7]. This AdoMet-dependent allosteric regulation is distinct for eukaryotic MTHFRs, as the regulatory domain thought to be responsible for AdoMet binding/sensing is not present in bacteria. However, to date, no structure of the inactive enzyme (T-state) exists, despite a relative wealth of biochemical data supporting the important role of AdoMet in regulating MTHFR activity[6–9,14,17]. Pioneering studies conducted in 1987 highlight that the eukaryotic MTHFR conformational ensemble exists in an R vs. T-state equilibrium, even as an apoenzyme ($K_{RT}$, ([T]/[R] in the absence of ligands = 0.30)[17]. AdoMet acts as an allosteric inhibitor, orchestrating and facilitating the structural transition to the inhibited conformation (T-state), while NADPH assumes the role of an activator, shifting the conformational equilibrium to the activated conformation (R-state)[17]. While AdoHcy does not act as an activator per se, it does act as a competitive binder for AdoMet[1]. However, the lack of a structural blueprint of eukaryotic MTHFR, particularly in its inactive, T-state configuration, prevents interrogating the underlying molecular mechanism that dictates the dynamics of the allosteric transition.

MTHFR dysfunction has been implicated in human diseased states[18,19] ranging from cancers[20] to psychiatric illnesses[21]. As such, providing a structural blueprint by which patient mutations in MTHFR can be understood has important implications for human health, and would allow for targeted drug discovery and therapy development. Drawing from the wealth of medical and clinical data on MTHFR, particularly regarding patient mutations, we aimed to decipher the allosteric regulation of MTHFR. However, the same features that enable MTHFR to bridge two metabolic cycles, namely its structural flexibility and dynamism, have also hindered attempts at structural characterization. Phosphorylation of human MTHFR is known to prime it for AdoMet inhibition, and uncoupling the contributions of phosphorylation from those of effector-induced allostery has proven difficult.

In this work, we use a thermophilic MTHFR homolog from *Chaetomium thermophilum* (*c*MTHFR) as a structural and biochemical model for *h*MTHFR. Although *c*MTHFR lacks the Ser-rich N-terminal phosphorylation region, the catalytic and regulatory domains of *c*MTHFR share high levels of sequence conservation/homology with those of *h*MTHFR (Fig. 1b, 38% sequence identity between domains), allowing us to investigate allosteric regulation independent of

phosphorylation status. Biochemical analysis of *c*MTHFR confirmed its NADPH preference and AdoMet-mediated inhibition, two features common to eukaryotic MTHFRs. Structural elucidation and recapitulation of the previously solved active, R-state conformation[7] displays the validity of our biochemical and structural model. Additionally, by introducing a human patient mutation, Arg357Cys (Arg315 in *c*MTHFR), we successfully crystallized *c*MTHFR in its inhibited, T-state, AdoMet-bound conformation. To complement and further understand the allosteric transition between each state, a detailed investigation of the phosphorylation sites in *h*MTHFR was conducted. Here, we discuss the possible shared mechanism of the allosteric transition of eukaryotic MTHFRs.

## Results

### Phosphorylation site and patient mutation analysis of *h*MTHFR

In eukaryotes, MTHFR is a homodimer consisting of catalytic and regulatory domains. Notably, the only structure of a eukaryotic MTHFR (*h*MTHFR, 6FCX)[7] shows that the dimer interface consists of interactions between the regulatory domains of two different protomers that form an antiparallel β-sheet dimer/β-sandwich. In this structure, AdoHcy was bound to the regulatory domain, and FAD was bound in the catalytic active site in a solvent-exposed manner, indicating that this captured *h*MTHFR structure likely represented a catalytically-competent, active enzyme in the R-state.

Human (*Homo sapiens*) MTHFR (*h*MTHFR) is post-translationally modified via phosphorylation of several residues, most of which reside on its N-terminus and the bulk of which are Ser. Phosphorylation of most sites is dependent on the initial phosphorylation of Thr34 in *h*MTHFR, which is thought to be the priming position for post-translational modification of *h*MTHFR by Pro-directed kinase(s)[22]. Phosphorylation plays a role in AdoMet-dependent allosteric regulation of MTHFR as indicated by experiments showing that mutation of the homologous Thr residue to Ala in other eukaryotic MTHFR proteins renders the enzyme less sensitive to AdoMet binding and inhibition[14,17]. However, the structural details of how the N-terminal phosphorylation sites affect the R- vs T-states conformational ensemble and how AdoMet binding leads to allosteric inhibition of *h*MTHFR are not known.

LC-MS/MS analysis of recombinant *h*MTHFR showed that, in our *h*MTHFR preparations, eleven amino acids were phosphorylated (Ser21, Ser23, Ser25, Ser26, Ser29, Ser30, Ser33, Thr34, Ser206, Ser394, and Ser412), eight of which overlapped with a previous report and three of which are identified here (Ser33, Ser206, Ser412). The previous report found 16 total amino acids were phosphorylated, 11 of which were found on the N-terminus[7]. The identification of Ser33 as one of the phosphorylation sites in the N-terminal Ser/Thr rich region, not previously reported, is noteworthy. Previously, Ser33 was postulated to play a role in phosphorylation based on data that showed that a Ser33Ala mutation impedes phosphorylation of other confirmed phosphorylation sites in *h*MTHFR[14], suggesting the importance of Ser33 phosphorylation in the early stages of sequential phosphorylations of *h*MTHFR. Our analysis reveals three phosphorylation sites outside the N-terminal Ser/Thr-rich region (Fig. 1c, Supplementary Figs. 1 and 2, Supplementary Discussion 1). While phosphorylation of Ser394 was previously established, Ser206 and Ser412 emerge as distinct findings. The functional implications of phosphorylation at Ser206, Ser394, and Ser412 and their role in modulating *h*MTHFR activity remain unexplored.

Despite the discovery of three phosphorylation sites, *h*MTHFR proved difficult to study structurally and recalcitrant to crystallization in our hands. Encouraged by the recapitulation of the results of previous phosphorylation studies, and taking a cue from patient mutations in *h*MTHFR[23], we set out to see if we could rationally trap the elusive T-state. Using the NADPH-menadione oxidoreductase assay, the effects of several patient mutations on activity and FAD release

were analyzed and compared to wild-type *h*MTHFR (Supplementary Figs. 3 and 4, Supplementary Discussion 2) to discern any trends that could provide insights into the molecular mechanism behind the conformational rearrangements influenced by AdoMet binding. Previous work on patients with MTHFR deficiencies has identified and characterized mutations that affect the activity and thermolability of human MTHFR, the latter of which is associated with FAD release[24].

Of all the patient mutations analyzed, Arg357Cys stood out due to its near abolishment of activity (14% relative to wild-type), without any alteration in FAD retention, as compared to the wild-type *h*MTHFR (Supplementary Figs. 3 and 4, Supplementary Discussion 3). The Arg357Cys patient mutant is based on a rare mutation (1081 C > T, Arg357Cys)[24]. Previous studies in *h*MTHFR have shown that AdoMet binding leads to a corresponding decrease in FAD release[25], indicating that the biochemical and, by extension, structural properties of the Arg357Cys mutant follow an analogous trend to those of the AdoMet-bound and inhibited enzyme in the T-state. In all, these results offer the tantalizing possibility that Arg357Cys favors the elusive inactive T-state form, mimicking the same conformation induced by AdoMet binding.

Thus far, despite numerous attempts, we have been unable to obtain structures of *h*MTHFR in its active or inactive conformations. However, given the promising biochemical data indicating that the T-state could be rationally trapped, we pursued alternate biochemical and structural models of *h*MTHFR, among other homologs. We intended to elucidate the structural mechanism behind AdoMet-mediated allosteric inhibition while granting greater insights into the intricacies involved in the conversion of chemical signals into structural rearrangements. Consequently, we explored an MTHFR homolog from *Chaetomium thermophilum* (*c*MTHFR), a thermophilic fungus, as a structural and biochemical model for eukaryotic MTHFRs. Although *c*MTHFR lacks an N-terminal phosphorylation site, the catalytic and regulatory domains of *c*MTHFR share reasonable homology with those of *h*MTHFR (Fig. 1b, 38% sequence identity). Moreover, the lack of an N-terminal phosphorylation region in *c*MTHFR was a desired feature, selected to facilitate crystallization through increased homogeneity, uncoupling phosphorylation status from effector-induced allosteric inhibition, and structural transitions.

### *c*MTHFR as a biochemical model for eukaryotic MTHFRs

We first set out to validate *c*MTHR as a functional biochemical model for *h*MTHFR and eukaryotic MTHFRs in general. We determined its reductant preference using the NAD(P)H-menadione oxidoreductase assay (Fig. 2a). *c*MTHFR$^{wt}$ can oxidize both NADPH and NADH but prefers NADPH (120 µM/min) over NADH (44 µM/min), akin to previous studies of eukaryotic MTHFR models[1,7,26]. The addition of exogenous FAD increases activity (150 µM/min), rescuing the loss of activity associated with FAD release while in solution. Having confirmed the NADPH preference of *c*MTHFR$^{wt}$, we set out to determine the effect of AdoMet on *c*MTHFR activity, again using the NAD(P)H-menadione oxidoreductase assay (Fig. 2a). In the presence of AdoMet (100 µM), activity was inhibited by 25%.

The Arg357Cys patient mutation was translated into *c*MTHFR, corresponding to the *c*MTHFR$^{R315C}$ mutant, and its NAD(P)H-preference and AdoMet-mediated inhibition were likewise interrogated. At the same enzyme concentration and in the absence of AdoMet, the Arg315Cys mutant displayed 14% relative activity compared to the wild-type enzyme (Fig. 2b), indicating that it existed in a primarily inhibited state.

The binding affinity of *c*MTHFR$^{wt}$ titrated with AdoMet was monitored via spectral changes using UV-Vis (Fig. 2c). The UV-Vis spectrum of *c*MTHFR$^{wt}$ is noticeably altered upon AdoMet binding, with FAD undergoing an AdoMet concentration-dependent absorbance quench and red-shift at 450 nm, altering the absorbance maxima of FAD from 453 nm to 463 nm. The sigmoidal nature of the FAD quench was assessed with a Hill plot, yielding a $K_{d,app}$ of ~18.8 µM, and a

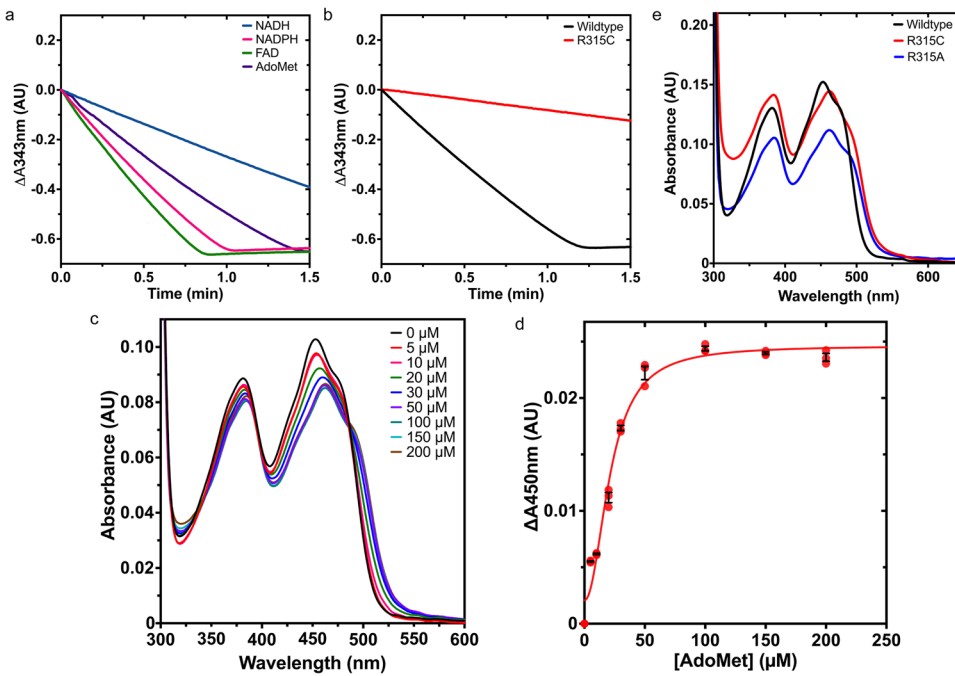

**Fig. 2 | Initial biochemical characterization of cMTHFR. a** NADPH/NADH preference and AdoMet-dependent inhibition of cMTHFR was assessed using the NADPH:menadione oxidoreductase assay. A concentration of 50 nM of cMTHFR was used. Consumption of NADPH or NADH (100 μM) was monitored by absorbance at 343 nm at room temperature. FAD (2 μM) and AdoMet (100 μM) were added separately. **b** NADPH-menadione oxidoreductase cMTHFR^R315C activity assayed in the same conditions as in **a**. Data was measured using $n = 2$ biologically independent samples. **c** UV-Vis spectral changes of cMTHFR^wt (10 μM) titrated with varying concentrations of AdoMet (0−200 μM). The typical flavin spectrum of

cMTHFR without AdoMet has two peaks at 370 and 450 nm. When AdoMet is added, these peaks are red-shifted, and the absorbance at 450 nm is decreased. **d** The absorbance changes at 450 nm of cMTHFR^wt (10 μM) as a function of AdoMet were fit with the Hill equation. **e** UV-Vis spectra of cMTHFR^wt, cMTHFR^R315C, and cMTHFR^R315A in the absence of AdoMet. Data in **c** are of a representative experiment, which has been repeated ≥3 times, with the corresponding curve in **d** obtained from $n = 3$ biologically independent samples analyzed using the mean ± SEM. Source data are provided as a Source Data file.

Hill coefficient of 2. These absorbance changes are likely due to changes in the FAD microenvironment upon AdoMet binding (Fig. 2d). Spectra of cMTHFR^wt and cMTHFR^R315C in the absence of AdoMet were compared (Fig. 2e). The spectrum of cMTHFR^R315C is similar to that of cMTHFR^wt with AdoMet bound (Fig. 2e) (maxima 462 nm); therefore, we posit that the Arg315Cys mutation results in a "locked" MTHFR conformation that is similar to that induced by the AdoMet binding, indicating that the cMTHFR^R315C mutant results in a conformation that resembles the elusive T-state. Encouraged by the biochemical results for both the cMTHFR^wt and cMTHFR^R315C mutant, we decided to test the validity of cMTHFR as a structural model, initially attempting to capture and recapitulate the R-state structure observed for hMTHFR.

### cMTHFR as a structural model for eukaryotic MTHFRs—R-State
cMTHFR^wt proved difficult to crystallize. As such, a mutant (cMTHFR^E21Q, L393M, V516F) was used instead, along with reductive methylation of surface-exposed lysine residues to aid in crystallization[27,28]. This construct was designed primarily to prevent adventitious AdoMet binding, which has been observed to copurify with MTHFR[7]. The construct was successfully crystallized, and its structure was solved to 3.4 Å (Fig. 3a) in the active R-state. Electron density for FAD was observed in every catalytic active site. Although electron density consistent with AdoHcy was found in the regulatory domain, it was weak and could not be faithfully modeled. Even so, the homodimer assembly was found to display the same dimer interface mediated by the regulatory domain, a β-sheet sandwich consistent with the one previously observed for AdoHcy-bound hMTHFR (R-state)[7] (Fig. 3a, c). Unlike bacterial MTHFR homologs, which use their catalytic domains for oligomeric assembly, cMTHFR, like hMTHFR, relies on its regulatory domain for oligomerization. The catalytic domains were also

found to face "away" from the dimer interface and each other and are not interacting with one another, in stark contrast to the direct role they play in dimerization in bacterial MTHFRs. In this active R-state, the si-face of FAD is unoccluded in the active site and forms an extended hydrogen-bonding network between N5 of FAD centered on conserved residues His81 and Asp50 (His127 and Asp92 in hMTHFR), along with a strong hydrogen bond between the universally conserved Thr52 (Thr94 in hMTHFR) and O4 of FAD (Supplementary Fig. 5). Given that the active site is solvent exposed and that access to FAD is unoccluded, the captured structure likely represents a catalytically-competent form of the enzyme in the active R-state, much like that observed for hMTHFR. Indeed, PISA analysis shows that while ~75.95% of the FAD active site binding interface is buried, a channel/tunnel exists, accessible to the bulk solvent, close to the adenine moiety, and with direct access to the si-face of FAD.

### Capturing the elusive T-state configuration using rational protein engineering
MTHFR conformational changes have been previously analyzed by limited tryptic digestion[8,14]. Matthews et al.[26] previously demonstrated that porcine liver MTHFR (molecular weight of 77 kDa) undergoes cleavage into two major fragments of 39 kDa and 36 kDa, facilitated by limited proteolysis using trypsin. Conformational changes of cMTHFR^wt, in the presence or absence of AdoMet, were visualized by limited proteolysis followed by SDS-PAGE (Supplementary Fig. 6). cMTHFR^wt exhibits increased susceptibility to trypsin in the presence of AdoMet; this suggests that at least part of the linker, nestled between the catalytic and regulatory domains as evident in the R-state (Supplementary Fig. 6a), becomes solvent exposed in the T-state. The differences in the protease accessibility of the linker can be possibly

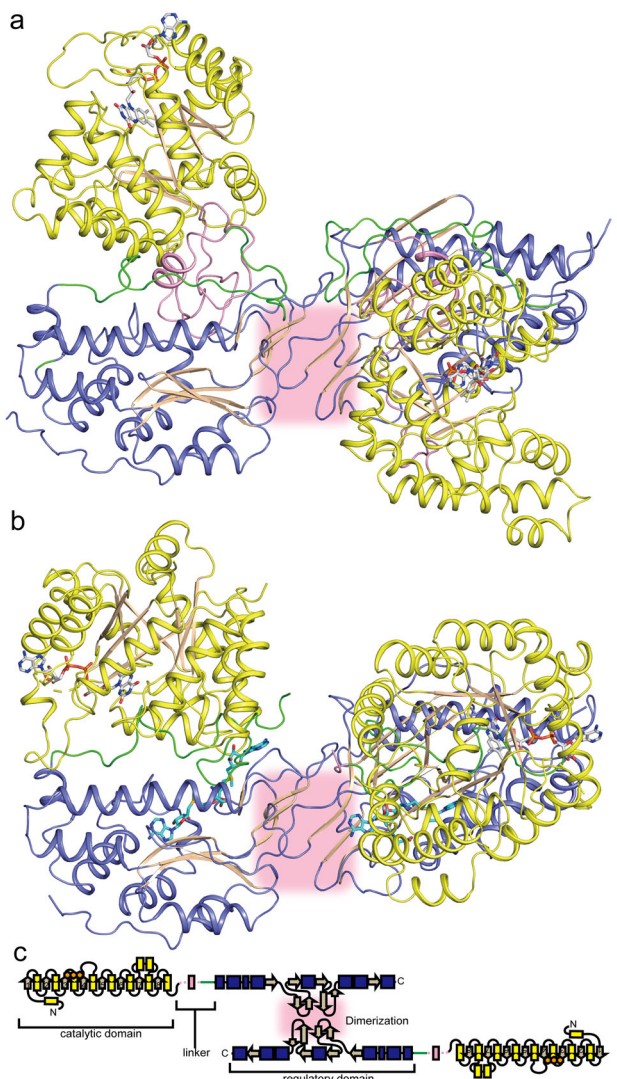

**Fig. 3 | Dimer structure of *c*MTHFR. a** Eukaryotic MTHFR structure in the R-state is illustrated in the ribbon diagram. The catalytic domain and regulatory domain are colored yellow and blue, respectively. The linker's N-terminal region is shown in pink, while the C-terminal one is shown in green. The dimer interface is highlighted with a red box. Bound FAD is shown in gray. No AdoHcy was found in our structure. **b** Eukaryotic MTHFR structure in the T-state is shown in the same orientation and using the same color scheme as that in the R-state (**a**). Bound AdoMet is shown in cyan. **c** Schematic diagram of the secondary structural features in the dimeric eukaryotic MTHFR structures.

attributed to an expected steric clash of Arg326 on the N-terminal region of the linker with the methyl group of AdoMet upon its binding (Ala368 in *h*MTHFR)[7]. This and possible local changes in the electrostatics could trigger linker rearrangement and transition to the T-state, where part of the linker becomes accessible. However, while the rich biochemical data available indicate that MTHFR exists as a conformational ensemble with distinct allosteric properties that do not ascribe to typical models of allostery (such as the fast-lag phase of AdoMet inhibition and biphasic inhibition nature)[1], the structural transition between the uninhibited (R-state) and AdoMet-bound inhibited (T-state) remains the subject of speculation given the lack of a structure in the T-state.

Encouraged by the recapitulation of the previously captured *h*MTHFR R-state structure, we set out to capture the elusive T-state structure. Taking a cue from our biochemical analysis of *h*MTHFR patient mutations, we set out to see if we could rationally trap the

T-state using the Arg357Cys patient mutant (Arg315Cys in *c*MTHFR). As compared to the other patient mutants analyzed, the combination of FAD retention and catalytic activity abolishment (Supplementary Figs. 3 and 4, 8% activity relative to wild-type), along with biochemical properties that mimic that of the AdoMet-bound state (Fig. 2c, e), led us to focus on trapping the T-state via mimicry of the Arg357Cys patient mutant. The analogous Arg315Ala mutant was used for structural studies to avoid the possibility of inter-disulfide bonds introduced by a Cys mutation. *c*MTHFR^R315A displays the same FAD absorbance quench and red-shift at 450 nm displayed by R315C (Fig. 2e). In addition, limited proteolysis of *c*MTHFR^R315A exhibits the same susceptibility to trypsin regardless of the presence or absence of AdoMet (Supplementary Fig. 7, Supplementary Discussion 3); in other words, the *c*MTHFR^R315A mutant exhibits effects that are AdoMet-independent, suggesting it is already preferentially trapped in the T-state in solution.

The *c*MTHFR^R315A mutant was successfully crystallized, and its structure solved to 2.8 Å (Fig. 3b). In this inactive T-state, the *si*-face of FAD is occluded by Tyr361 on the C-terminal region of the linker, which forms a π-stacking interaction (Supplementary Fig. 8). Additionally, each monomer adopts a more compact structure relative to the R-state, where the catalytic domain is rotated by 45° coupled to a translation of ~16 Å, burying the catalytic domain between the linker and the regulatory domain. PISA analysis shows that ~95.53% of the FAD active site binding interface is buried, aided by the interaction from the linker. The regulatory domain was found to bind AdoMet, as expected for the T-state. Unexpectedly, two molecules of AdoMet were found per regulatory domain. One is found bound in the same site that AdoHcy occupies in the R-state (AdoMet site-1), while the other is found in a second adjacent site (AdoMet site-2) that was occluded by residues of the linker (aa301–305) in the R-state structure (Supplementary Figs. 9 and 10).

## Linker rearrangement dictates and defines R- to T-state transition

The overall structure of the *c*MTHFR^R315A mutant is shown in Fig. 3b. The dimer interface of *c*MTHFR^R315A in the T-state is similar to that of the *h*MTHFR structure (6FCX) and *c*MTHFR in the R-state, mediated by the same β-sandwich formation between two monomers. Notably, the catalytic domains remain free of direct interdomain contact within the dimeric assembly. The superposition of *c*MTHFR^R315A in the inactive T-state with the *c*MTHFR structure in the active R-state using the regulatory domain as the reference point is shown in Fig. 4a. While the catalytic and regulatory domains of each state are indistinguishable (Supplementary Fig. 11), the global rearrangement between states highlights that the linker plays a focal role in channeling the R- to T-state transition. Indeed, the linker is found to be more flexible in the T-state and partially solvent-exposed, with part of the N-terminal region unable to be modeled.

The individual domains were compared via structural alignment/superposition. Structural alignment of the regulatory domains of the R and T-states show they are nearly identical (RMSD ~ 0.39 Å, Supplementary Fig. 11a, Supplementary Discussion 4), highlighting a conserved topology. The β-sandwich dimer interface is composed of adjacent regulatory domains in both configurations. Structural alignment of the catalytic domains likewise demonstrates that the individual domains are indistinguishable between the two different states (RMSD ~ 0.53 Å, Supplementary Fig. 11b, Supplementary Discussion 4), highlighting their structural rigidity. Even so, there are pertinent global changes: in the T-state, for instance, the catalytic domains re-orient with respect to the regulatory domain, and MTHFR adopts a more compact conformation overall in this inactive state. Consequently, the transition between the R- to the T-state leads to an overall contraction of the global structure that shifts the catalytic domains (and their FAD centers) ~23 Å closer to one another.

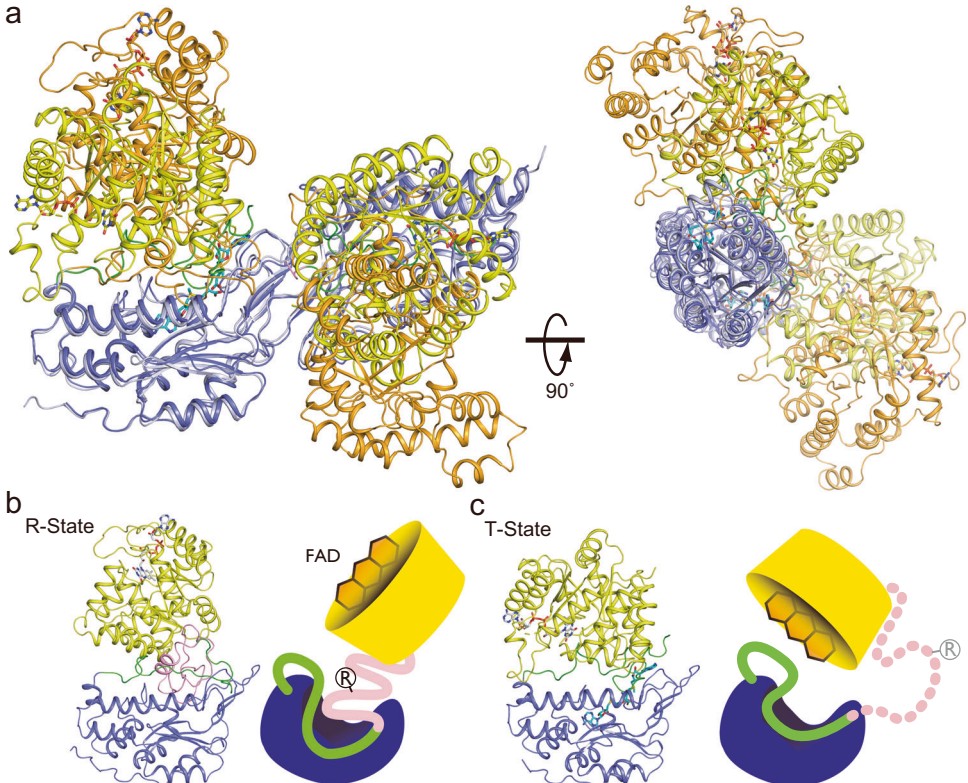

**Fig. 4 | Comparison of *c*MTHFR in R- and T-state. a** Dimer comparison of the R and T-states. The catalytic domain and regulatory domain are colored yellow and blue, respectively, for the T-state, and light orange and light cyan, respectively, for the R-state. FAD is colored in gray, and AdoMet bound to the T-state is colored in cyan. **b** Monomer in the R-state illustrated in **b** and monomer in the T-state in **c** using the same color scheme as in Fig. 3; these are also drawn in cartoon style. Since the structure in T-state lacks the N-terminal part of the linker (pink), the region is drawn as a dotted line in the cartoon. In the R-state, MTHFR accommodates the N-terminal pink region of the linker between the catalytic and regulatory domains. The C-terminal portion of the linker faces away from the active site, resulting in substrates binding to the active site of MTHFR. The C-terminal linker portion is herein called the velcro-wedge region (green). The circled R shows the Arg residue mutated to obtain the "T-state-lock" MTHFR, which is on the N-terminal linker portion, herein called the retractable-hinge region (pink). The R and T MTHFR states are illustrated in a schematic mode. Cartoons in **b**, **c** were created with BioRender.com released under a Creative Commons Attribution-NonCommercial-NoDerivs 4.0 International license.

The catalytic domain transitions relative to the regulatory domain, from R- to the T-state, are primarily driven by the rearrangement of the linker. This flexible linker (aa293–370) consists of two regions, an N-terminal retractable-hinge (aa293–348, pink, Figs. 4–7) and a C-terminal velcro-wedge (aa349–370, green, Figs. 4–7), so called because it forms a distinct interface between the catalytic and regulatory domains (velcro) and occludes access to FAD via Tyr361 (wedge). The linker is arguably where the most drastic structural changes are observed (Figs. 4b, 5a, and 6a). The active site, particularly the FAD cofactor, becomes buried within the catalytic domain due to the rearrangement of the linker, with the velcro-wedge region embedded between the regulatory and catalytic domains, forming a distinct interface. The N-terminal retractable-hinge region undergoes a helix-loop transition between the R- and T-states, serving as a hinge that guides and governs the observed global structural changes. The most prominent structural feature of the retractable-hinge region in the R-state is the presence of an α-helix (aa329–337, Figs. 4b, 5a, and 6a); most of the retractable-hinge region could not be modeled (aa296–336) in the T-state, indicating increased flexibility and/or increased solvent exposure when compared to the R-state, where it was embedded between the catalytic and C-terminal velcro-wedge region. Our structures identified hotspots (near Arg315 in *c*MTHFR) in structural regions of known allosteric importance (Fig. 5). When the linker between the catalytic and regulatory domains is retracted, Arg315 can interact with Glu318 and Thr336 in this active state (Fig. 5a, b). Thus, loss of Arg315 in the patient mutant can lead to

destabilization of the retractable-hinge region and, in turn, stabilization of the T-state. Arg315 could thus be a critical residue in stabilizing the R-state: there are five positively charged or polar residues near Arg315 (Ser310, Arg316, Arg321, Asn329, and Arg335) that form a positively charged patch. The positively charged patch is conserved in vertebrate MTHFR (Fig. 5c, Supplementary Figs. 12 and 13).

The N-terminal linker region can be thought of as a hinge, whereby its extension causes a corresponding contraction/compacting of the catalytic domain towards the regulatory domain, yielding the T-state configuration. Indeed, this region occludes part of the allosteric site in the R-state (aa303–306, HALP). Additionally, the C-terminal velcro-wedge region serves to form part of the allosteric binding pocket (aa339–347) in the T-state structure, with aa395–409 of the regulatory domain acting as a lid over the allosteric site (Fig. 7). As such, the N-terminal linker region can be thought to act as a wedge in the R-state, whereby its occlusion of the secondary AdoMet binding site favors the R-state conformation.

The C-terminal velcro-wedge region, previously solvent exposed in the R-state, becomes completely embedded in an interface between the catalytic and regulatory domains. Tyr361 on the velcro-wedge region repositions a dramatic ~16 Å, forming a π-stacking interaction with FAD, directly occluding its *si*-face (Fig. 6b). The combined effect of these changes explains the catalytically inhibited nature of the T-state (Fig. 6b), where the Tyr361 and the C-terminal linker (Tyr finger) act as an autoinhibitory element, a form of intrasteric (active site-directed) regulation[29,30]. The movement of this loop/linker effectively

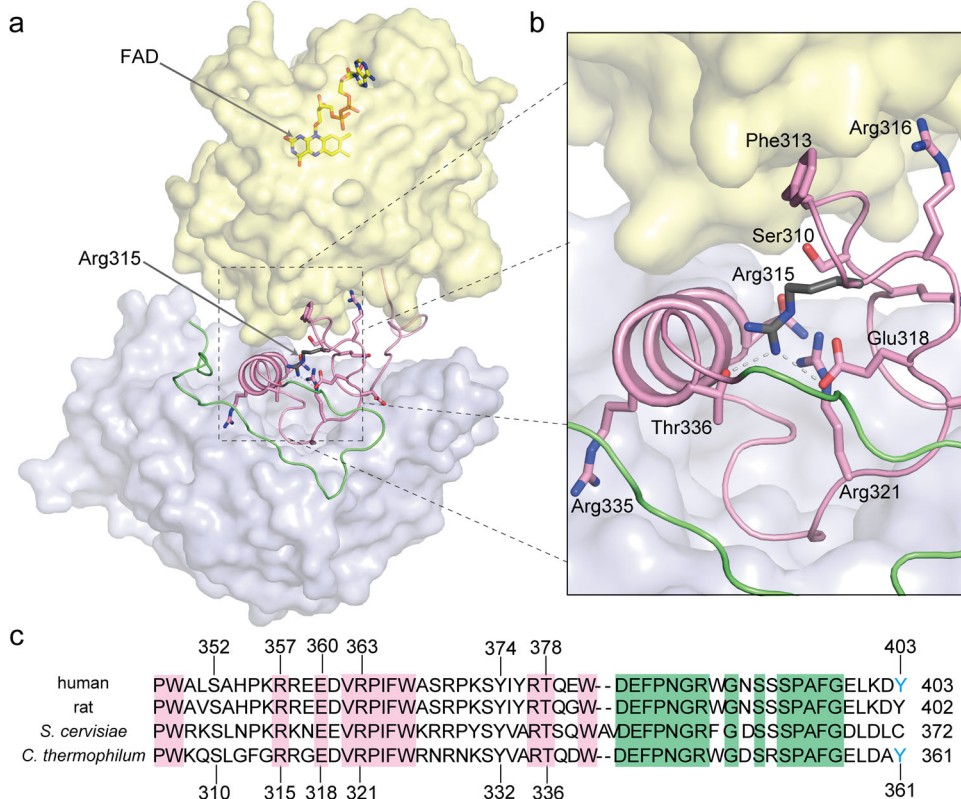

**Fig. 5 | The role of the linker in interdomain interactions in the R-state.**
**a** Surface depiction of the R-state shown using the same coloring scheme as in
Fig. 4 highlighting the position of Arg315. The Arg315Cys mutation is critical to
destabilize the R-state or to stabilize the T-state because the mutation yields T-
state-locked cMTHFR. **b** The retractable-hinge mediates linker interactions
between domains, ostensibly crucial for the allosteric transition. Arg315 interacts
with Thr336 and Glu318, which could stabilize the R-state, indicating that loss of
the positively charged side chain by Cys mutation would destabilize the R-state.

Moreover, there are five positively charged or polar side chains, Ser310, Arg316,
Arg321, Asn329, and Arg335 found near Arg315. Given that the N-terminal phos-
phorylation site is close to the retractable-hinge region, the negatively charged
phosphorylation sites would attract those positively charged side chains. **c** Amino
acid alignment of MTHFR linker from *Homo sapiens*, *Rattus norvegicus*, *Sacchar-
omyces cerevisiae*, and *Chaetomium thermophilum*. Conserved amino acids are
highlighted in pink and green for the retractable-hinge and velcro-wedge regions,
respectively.

serves to block access to the catalytic domain, trapping the enzyme in
an inactive state while providing a closed pocket and stabilizing
interaction with FAD that simultaneously explains the abolished cata-
lytic activity and retention of the FAD cofactor observed with the
R357C patient mutant (R315C/A in cMTHFR); it also explains the altered
UV-Vis spectrum observed upon AdoMet binding (Fig. 2c, e), which
could thus be attributed to the altered FAD microenvironment[31] in the
T-state configuration. As such, the C-terminal linker region can, in turn,
be thought to act as a lock/wedge in the T-state, whereby its interac-
tion with the *si*-face of FAD, protrusion into the catalytic domain, and
(re)positioning between the catalytic and regulatory domains serve to
stabilize and favor the T-state.

**Binding of two AdoMet molecules promotes drastic linker
rearrangements**
The regulatory domain was found to bind AdoMet, as expected for the
T-state. Unexpectedly, two molecules of AdoMet were found per reg-
ulatory domain. One AdoMet molecule is found in the same site that
AdoHcy occupies in the R-state (site-1), while a second AdoMet mole-
cule is bound in a second adjacent site (site-2) that was occluded by
residues of the retractable-hinge region (aa301–305, Fig. 7, Supple-
mentary Figs. 9 and 10) and whose pocket is in part lined by the velcro-
wedge region (aa339–347, Fig. 7a). Although the second AdoMet
binding site is hidden in the R-state, the large conformational change
that defines the T-state allows for the accommodation of two AdoMet
molecules per monomer. AdoMet binding to site-1 and a steric clash

with Arg326 on the retractable-hinge region or Ala408 (structurally
analogous to Ala368 and Ala461 in hMTHFR, respectively) could pre-
cipitate the initial helix-loop transition, whose extension corresponds
to movement of the catalytic domain.

Previous work identified residues important to AdoMet binding,
finding their sequence to be aa373-SYIYRTQEWDEFPNGRWGNS-392 in
hMTHFR. In the crystal structure of cMTHFR^R315A, the peptide aa338-
DWDEFPNGRWGDSRSPAFGELDA-360 occupies the corresponding
position. While the AdoHcy-binding site (AdoMet site-1) is separated
from this peptide, it is close to the second AdoMet binding site
(AdoMet site-2), suggesting that the previous study might have
revealed the location of the second, cryptic AdoMet-binding pocket
(AdoMet site-2)[32]. Another study identified residues important to
AdoMet binding, where residues in AdoMet site-2 were surprisingly
found to be important for AdoMet binding as a whole, despite them
corresponding to the linker and not the regulatory domain[33]. The
unveiling of a cryptic AdoMet site previously occupied by aa297-
DRPLKHALPW-307 of the retractable-hinge region of the linker in the
R-state to T-state transition helps explain this previously puzzling
finding[32].

MTHFR is an allosteric enzyme that undergoes slow, hysteretic
changes in activity in response to its physiological regulator, AdoMet
(lag)[1,17]. This was initially ascribed to the timing of retraction/retention
of the hinge region upon AdoMet binding to the R-state, where the
triggered conformational change represents the rate-limiting step
between the transition from the R-state to the T-state, followed by a

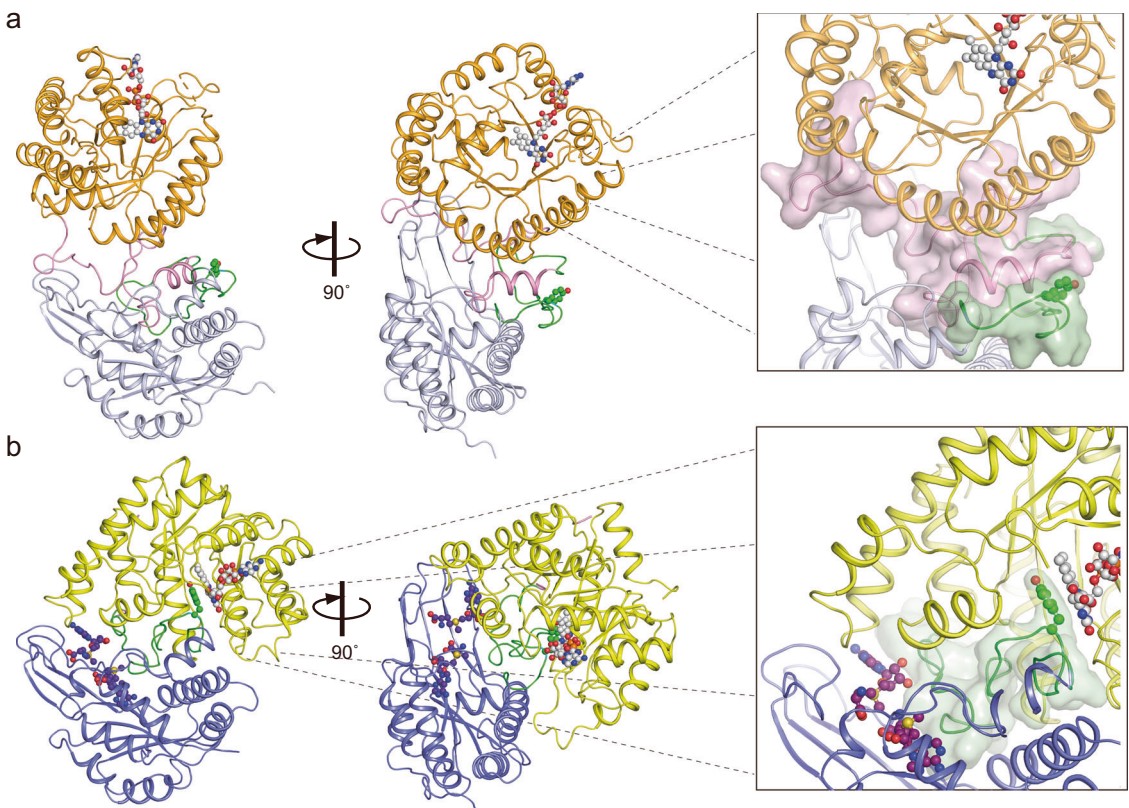

**Fig. 6 | Linker region occludes FAD in MTHFR T-state.** Structures of *c*MTHFR in R-state (**a**) and T-state (**b**) are shown using the same coloring scheme as in Fig. 4. Tyr361, present on the C-terminal portion of the linker (green), occludes the *si*-face of FAD in the T-state (bottom-right, inset), while that portion of the linker remains solvent-exposed in the R-state (top-right, inset).

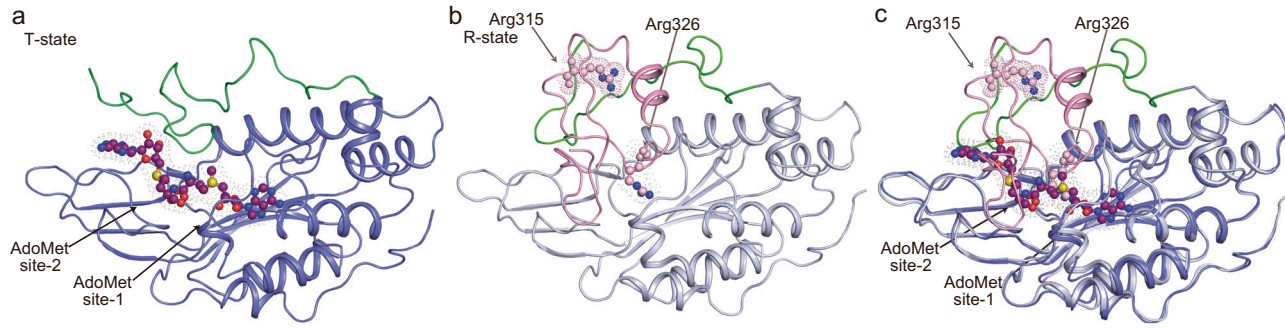

**Fig. 7 | Effector molecule binding site comparison suggests a possible mechanism for how MTHFR transitions from the R to T-state is triggered.** Structures of regulatory domain of *c*MTHFR in T-state (**a**) and R-state (**b**) Two AdoMet binding sites in T-state are compromised by the regulatory domain, the inhibitory region of the linker, and the catalytic domain. The catalytic domain does not directly make contact with the regulatory domain in the R-state (**b**) because of the presence of the retractable region of the linker. To demonstrate the effect of AdoMet binding on MTHFR, the regulatory domain and the retractable region of the linker are superimposed (**c**). AdoMet and AdoHcy share the AdoMet site-1 (**c**). As AdoMet binds to the AdoMet site-1, the *S*-methyl moiety of AdoMet causes steric clash with Arg326 (**c**). The steric clash induces the repositioning of the retractable region and disfavors the R-state. Repositioning of the retractable region reveals the cryptic AdoMet site-2 that facilitates the formation of the T-state.

slow "isomerization" towards the final, inhibited form (T-state). However, the authors did acknowledge the possibility that the biphasic nature of AdoMet inhibition could be explained by multiple AdoMet binding events, an initial "burst" phase for the primary AdoMet binding event (R-state, site-1), and a secondary AdoMet binding event to an intermediary configuration that hastens the allosteric transition (I to T-state, site-2)[17]. While this manuscript was under review, Froese et al.[34]. were able to obtain a human MTHFR structure in its inhibited state (T-state). Our spectrophotometric data indicate that the observation of two molecules of AdoMet in our T-state structure is not an artifact, and that we do indeed observe multiple AdoMet binding events in solution. Froese et al.'s results confirm our findings, and together, they complement each other, finding that two molecules of AdoMet are bound in their T-state structure and that restructuring of the linker plays a pivotal role in the R- to T-state transition[34].

The key to understanding previous biochemical data[1,17] was the adoption of an effector-induced pathway model wherein binding of one AdoMet molecule to the R-state triggers a conformational change

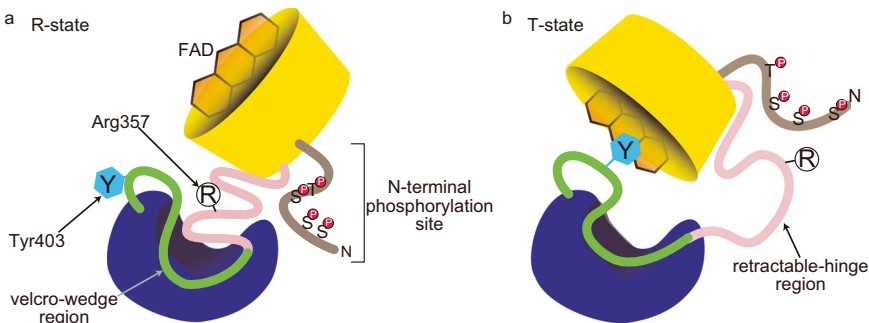

**Fig. 8 | A proposed mechanism for how phosphorylation of *h*MTHFR enhances the AdoMet-sensitivity. a** The N-terminal phosphorylation site (brown) is near the retractable region of the linker (pink), which is buried in the R-state. **b** In the T-state, the phosphorylation site and the retractable region may interact, although the retractable region would be flexible and solvent exposed. Cartoons in **a** and **b** were created with BioRender.com and released under a Creative Commons Attribution-NonCommercial-NoDerivs 4.0 International license.

that favors and allows for binding of a second AdoMet molecule to this intermediary state (hypothetical I-state), providing the committed step towards the T-state, with cooperative AdoMet binding[17]. Indeed, our spectrophotometric titration of AdoMet to *c*MTHFR[wt] (Fig. 2d) demonstrated the sigmoidal nature of the FAD quench, best fit with a Hill plot with a Hill coefficient of 2. This indicates that AdoMet binding is cooperative and that there are thus multiple AdoMet binding events, the first of which facilitates the binding of the second AdoMet molecule. The "slow" step observed and first reported by the authors is thus best explained by the second of their two hypotheses: that there are two AdoMet binding events, and that the binding of the first AdoMet molecule triggers the R-T-state transition by cooperatively enabling the binding of a second AdoMet molecule.

While the regulatory domain remains relatively static between both captured configurations, aa318–326 of the linker (retractable-hinge region) interacts with the dimer interface in the R-state, while aa296–336 overall becomes solvent-exposed and flexible in the T-state. The linker serves a dual role as a sensor and hinge, as the N-terminal retractable-hinge region (aa293–348) undergoes a helix-loop transition, extending and unwinding in the R- to T-state transition (Figs. 7 and 8). AdoMet binding to site-1 is the likely impetus for the initial structural changes that precipitate the transition, as site-2 is only revealed in the T-state (i.e., after AdoMet site-1 is occupied). This implies that the initial effector-mediated remodeling restructures the regulatory domain as well, unveiling the cryptic AdoMet site and adopting a conformation that favors the binding of a second AdoMet molecule.

## Discussion

The regulation of eukaryotic MTHFR has proven to be multipronged and multifaceted, relying on phosphorylation status as well as effector-induced and stabilized conformational changes, all driven and achieved via the restructuring of its linker. Allostery via linker grants MTHFR a multimodal allosteric regulatory (cap)ability. The intricately layered role played by the linker, granted by its two disparate functional regions, indicates that MTHFR has evolved an original and, to our knowledge, unprecedented way of regulating its function via effector-mediated changes. Using a linker, eukaryotic MTHFR has shown that it can channel an initial chemical event, such as AdoMet binding, into a complex mechanical rearrangement whereby it orchestrates a drastic global contraction of the dimer, organizing a coupled rotation and translation of the catalytic domain (45° and ~16 Å, respectively). In addition, it simultaneously reveals a secondary cryptic AdoMet site and occludes access to the *si*-face of FAD. Most astonishingly, a portion of the linker ends up being buried between the regulatory and catalytic domains, though it still participates in AdoMet site-2, FAD, and T-state stabilization. The transition from the R- to

T-state involves a drastic decrease in the surface area of the overall structure, with a concurrent contraction reminiscent of hydrophobic collapse. The velcro-wedge region of the linker is solvent exposed in the R-state but becomes completely buried in the T-state; the opposite is true for the retractable-hinge region, where its environment is hydrophobic in the R-state but becomes solvent exposed in the T-state.

It is worth mentioning that phosphorylation itself must be considered to play an important role, especially in *h*MTHFR, where it has been shown that phosphorylation of predominantly N-terminal residues increases sensitivity to AdoMet-dependent allosteric inhibition. The retractable-hinge region of the linker is flexible, particularly in the T-state, and as such, it can presumably interact with the phosphorylated amino acid side chains of the N-terminal (Fig. 8). The phosphorylation status of MTHFR adds yet another layer to the mechanism by which effector-mediated structural changes are channeled via the linker.

This effector-mediated allosteric conformational change is wholly unexpected, as is the unveiling of a cryptic secondary AdoMet binding site. AdoMet site-2 is only unveiled after AdoMet site-1 is occupied and the conformational rearrangement has begun. AdoMet site-2's direct proximity and interaction with the velcro-wedge region of the linker (F342) and the regulatory domain (W447) (Supplementary Fig. 10), along with its more solvent-exposed nature relative to site-1, make it a prime candidate site to initiate the reverse transition (T- to R-state). Therefore, signaling that AdoMet/AdoHcy concentrations in the cell require decreased flux into the folate cycle could be committing the enzyme to T- to the R-state transition. It has been shown that NADPH-binding precipitates the reverse transition (T- to R-state), eliciting the opposite effect of AdoMet binding[17]. However, how the T-state, once attained, can transition to and rearrange back to the R-state is not clear. In the R-state, when AdoMet binds to site-1, it is proximal to and clashes with the Arg326 retractable-hinge region of the linker, providing a direct method via which the initial AdoMet binding event to site-1 can trigger the observed conformational changes (Fig. 7c).

The linker region likely has additional roles to those mentioned above. The flexibility of the linker in the T-state in our experimental structures and computational models hints that it is also crucial in the T- to R-state transition. As it stands, while potential mechanisms to transition from the T- to the R-state are postulated, such as dephosphorylation of PTM residues, AdoHcy-binding and/or AdoMet-unbinding, and NADPH-binding, these remain solely theoretical, and further work must be done to ascertain what governs T- to R-state transition ("reactivation"). While it is prudent to assume that the linker will play a major role, the initial event that triggers the reconfiguration and the interactions the retractable-hinge and velcro-wedge regions could form are unclear.

The outsized importance of MTHFR as the convergence point and rate-limiting enzyme in one-carbon metabolism has forced Nature to adopt two common structural motifs (TIM barrel, β-sheet) and use unheralded features, in this case, a linker, to achieve a distinct form of allostery. Through a linker region, MTHFR can govern one-carbon metabolism flux, sense and react to cellular conditions (AdoMet/AdoHcy concentration and ratio), and use ~80 amino acids to sense, block, protect, and trigger dramatic global conformational changes. Though already complex, the structural and mechanistic details regarding MTHFR allostery are only just being described. Whether MTHFR's distinct take on allostery via the use of its linker is more common than previously appreciated remains unclear. The use of two allosteric inhibitor sites, one cryptic and only exposed after the initial inhibitor binding event, represents a distinct remodeling of an allosteric site; the unexpected stoichiometry of two AdoMet molecules bound per monomer indicates that each site plays a differing role in the allosteric regulation and subsequent structural transition between R- and T-states and that Nature has strategically tuned its ability to sense and respond to the AdoMet/AdoHcy ratio present in the cell, gating one-carbon flux accordingly.

## Methods

### Preparation of recombinant hMTHFR

Preparation of recombinant hMTHFR was performed using a baculovirus-insect cell expression system (Invitrogen, cat. 10359-016)[14,25]. The cDNA encoding MTHFR was sourced from *Homo sapiens* [GenBank accession code U09806]. Both the wild-type variant and the Arg357Cys mutant were obtained via the baculovirus-insect cell (Sf9) expression system (Invitrogen, cat. 11496-015). Generation of a donor vector for the Arg357Cys mutant was performed using the QuikChange mutagenesis kit (Agilent). Recombinant hMTHFR obtained from Sf9 cells was resuspended in 50 mM potassium phosphate buffer (KPB), pH 7.2, 0.1 M NaCl, and 1% Triton X-100 (8 mL per 1 g of pellet). The resuspended cell pellet was lysed via sonication (4 °C, 10 min). The crude lysate was centrifuged (20 min, $5000 \times g$, 4 °C), decanting the supernatant to remove any cellular debris/pellet. To this supernatant, 2 grams of DEAE cellulose (DE-52, Whatman) was added and stirred in an ice-water bath. The resulting gel slurry was then packed into an empty 2.5 cm diameter column. After a wash with 50 mM KPB, pH 7.2, and 0.1 M NaCl (50 mL), the protein was eluted in bulk with 50 mM KPB, pH 7.2, and 0.3 M NaCl (20 mL). The eluate was then applied onto a Ni-affinity column (His-Trap Chelating HP, Cytvia/GE, 1 mL) pre-equilibrated with 50 mM KPB, pH 7.2, 0.3 M NaCl, and 0.05 M imidazole. Following a wash (50 mM KPB, pH 7.2, 0.3 M NaCl, and 0.1 M imidazole), the protein was eluted in bulk (50 mM KPB, pH 7.2, 0.3 M NaCl, and 0.3 M imidazole). The eluted fraction was dialyzed at 4 °C overnight (50 mM KPB, pH 7.2). The resulting dialysate was concentrated and stored at −80 °C.

### Preparation of recombinant cMTHFR

The cDNA encoding MTHFR was sourced from *Chaetomium thermophilum var. thermophilum* DSM 1495, NCBI Gene Locus tag CTHT_0033700 [https://www.ncbi.nlm.nih.gov/gene/18257408], GenBank accession code XP_006693807 [https://www.ncbi.nlm.nih.gov/protein/XP_006693807.1]. *C. thermophilum* possesses two MTHFR genes with locus tags CTHT_0065570 and CTHT_0033700. The latter was selected for its greater sequence similarity to the *Saccharomyces cerevisiae* Met13 gene[35], whose activity is modulated by AdoMet, and was synthesized by GeneArt (Invitrogen). The wild-type gene, originally cloned in a pMA vector, was subcloned in a pMSCG7 vector using ligation-independent cloning (LIC). The expression vector was designated as pMCSG7(cMTHR^wt). To express the R315A mutant and E21Q, L393M, V516F triple mutant, site-directed mutagenesis was performed using the QuikChange

mutagenesis kit (Agilent) to construct pMCSG7(cMTHR^R315A) and pMCSG7(cMTHFR^E21Q, L393M, V516F). *BL21star(DE3)* was used for protein expression. A complete list of the bacterial strains, plasmids, and primers used in this study are listed in Supplementary Table 2. *E. coli* transformed with either pMCSG7(cMTHR^wt), pMCSG7(cMTHR^R315A), or pMCSG7(cMTHR^E21Q, L393M, V516F) was propagated at 37 °C in Luria Broth containing 50 μg/mL ampicillin, and protein overexpression was induced via auto-induction[36,37]. Cells were grown at 30 °C overnight before harvesting via centrifugation and stored at −80 °C.

The harvested cell pellet was resuspended in 50 mM KPB, pH 7.4 (4 mL per 1 g of pellet), to which lysozyme (0.1 mg/mL) and PMSF (1 mM) were added. The resuspended cell pellet was lysed via sonication (4 °C, 5 s on, 5 s off, 5 min total). The crude lysate was centrifuged (45 min, $20,000 \times g$, 4 °C), decanting the supernatant to remove any cellular debris/pellet. The crude supernatant was collected and loaded onto a Ni-affinity column (His-trap Chelating HP, Cytvia/GE, 5 mL) pre-equilibrated with 50 mM KPB, pH 7.4, and 20 mM imidazole. The column was washed using the equilibration buffer (20 mM imidazole), then 80 mM imidazole, and the protein was eluted in bulk with a buffer consisting of 50 mM KPB, pH 7.4, and 250 mM imidazole. Protein-containing fractions as judged by SDS-PAGE analysis were collected and subjected to a TEV digest, with dialysis at 4 °C overnight (50 mM KPB, pH 7.4). The dialysate was loaded onto a pre-equilibrated HiLoad 16/600 Superdex 200 pg gel filtration column (SEC buffer, 25 mM Tris, pH 7.4, 0.1 M KCl, and 1 mM TCEP). Fractions containing the desired protein, as judged by SDS-PAGE, were pooled and concentrated via centrifugation in 25 mM Tris, pH 7.4, 50 mM KCl, 1 mM TCEP to yield purified cMTHFR^wt/cMTHFR^E21Q, L393M, V516F (~20 mg/mL) or cMTHFR^R315A (~50 mg/mL), which was stored at 4 °C or flash-frozen for long-term storage at −80 °C.

### Biochemical analysis of hMTHFR patient mutations

The recombinant hMTHFR^R357C mutant was produced using the baculovirus-insect cell expression system[25]. Activity for the cell extract expressing histidine-tagged hMTHFR^R357C was assessed using the NADPH:mendione oxidoreductase assay. FAD release from hMTHFR was measured according to our previous method[25], with minor modifications. A spectrofluorophotometer RF-5300PC (Shimadzu) with a cell-temperature controller was used. The excitation and emission wavelengths were set at 390 nm and 525 nm, respectively. Concentrated hMTHFR was diluted directly into pre-warmed 50 mM KPB, pH 7.2 (3 mL) at 46 °C to a final concentration of 100 nM, and the fluorescence intensity from released FAD was monitored for 10 min.

### Enzyme assay and UV-Vis spectroscopy of cMTHFR

Enzyme assays using cMTHFR were performed using a previously established assay[2], with some minor modifications. NADPH:menadione oxidoreductase activity was measured using a Cary 100 Bio spectrophotometer (Agilent Technologies, Inc). The reaction mixture without menadione was prepared at room temperature, containing 50 nM of cMTHFR with 100 μM NADPH or NADH in 50 mM KPB, pH 7.2. Reactions were initiated by adding concentrated menadione solution (100 μL per mL) to the cuvette. The consumption of NADPH (or NADH) was monitored via absorbance at 343 nm at room temperature. The change in concentration of NAD(P)H was determined using the extinction coefficient of oxidized NAD(P)H of 6220 $M^{-1}$ $cm^{-1}$ at 343 nm. Once NADPH preference was established, the effect of S-adenosylmethionine (AdoMet) on activity was assessed by adding 100 μM of AdoMet to the reaction mixture before menadione addition. Similarly, to test the effect of FAD on activity, 2 μM of FAD was added to the reaction mixture before menadione addition. Comparison of the UV-VIS spectra of cMTHFR constructs, monitoring the changes at 450 nm associated with FAD, were assessed between cMTHFR^wt,

$c$MTHFR$^{R315C}$/$c$MTHFR$^{R315A}$ in the absence of exogenous AdoMet. To determine the changes in absorbance at 450 nm as a function of AdoMet concentration, spectra were recorded using 10 μM of $c$MTHFR$^{wt}$ and 0–200 μM of AdoMet.

## Surface lysine methylation

For $c$MTHFR crystallization in the R-state, purified protein ($c$MTHFR$^{E21Q, L393M, V516F}$) underwent treatment with formaldehyde and dimethylamine-borane complex (Sigma-Aldrich) for surface lysine methylation[27,28]. After lysine methylation, the protein was loaded onto a pre-equilibrated HiLoad 16/600 Superdex 200 pg gel filtration column (SEC buffer, 25 mM Tris, pH 7.4, 0.1 M KCl, and 1 mM TCEP). Fractions containing the desired protein, as judged by SDS-PAGE, were pooled and concentrated via centrifugation in 25 mM Tris, pH 7.4, 50 mM KCl, 1 mM TCEP to yield surface lysine-methylated $c$MTHFR$^{E21Q, L393M, V516F}$ (~20 mg/mL) which was stored at 4 °C or flash-frozen for long-term storage at −80 °C.

## Post-translational modification of recombinant $h$MTHFR

Phosphorylation sites on $h$MTHFR were analyzed according to a modified protocol[38,39]. Briefly, purified His-tagged recombinant wild-type $h$MTHFR (~50 μM, 300 μL, ~1.1 mg total) was treated with 10 mM DTT for 30 min at room temperature, followed by iodoacetamide (50 mM final concentration) for an additional 30 min at room temperature in the dark. Trypsin (20 μg, Pierce Trypsin protease, MS-Grade) (~1:28 ratio trypsin:$h$MTHFR) was used for digestion, and the mixture was incubated at 37 °C overnight. The reaction was quenched by adding 50% TFA (25 μL, final concentration, 0.5% v/v). Phospho-peptides were enriched using a TiO$_2$ column (GL Science) according to the manufacturer's instructions. Non-phosphopeptides were eluted using 80% AcN, 0.4% TFA. Phosphopeptides were eluted using 5% ammonium hydroxide and then concentrated using a C$_{18}$ tip column (ZipTip, Millipore) according to the manufacturer's instructions. Samples were, vacuumed, dried, and resuspended in 60% AcN, 0.1% FA prior to LC-MS/MS analysis.

Phosphopeptide-enriched tryptic fragments were analyzed using a Thermo Scientific Orbitrap Fusion Lumos Tribrid MS coupled with the UltiMate 3000 RSLCnano liquid chromatography system (University of Michigan, Department of Chemistry). The mobile phase consisted of Buffer B (0.1% FA in 80% AcN) and Buffer C (0.05% TFA in 2% AcN). Peptides were eluted using a gradient with increasing concentrations of buffer B over 40 min at a flow rate of 300 nL/min (total run time 60 min). MS data were acquired in data-dependent mode (mass range: 355–1700, resolution: 120,000, target: $4 \times 10^5$, maximum injection time: 50 ms) followed by MS/MS via higher energy collision dissociation (HCD; normalized collision energy: 38%, resolution: 60,000 in profile mode, target value: 50,000, maximum injection time: 118 ms, isolation window: 0.7 m/z). Precursor ions of unassigned or +1 charge state were rejected. Additionally, precursor ions already isolated for fragmentation were dynamically excluded for 60 s. The proteomics data analysis was performed using an automated workflow on Proteome Discoverer 2.2.0.388 (Thermo Scientific) using the SEQUEST database search algorithm. Mass spectra were searched against a target-decoy database consisting of the forward and reverse sequences of the UniProt human reference proteome. Carbamidomethylation of cysteine was set as a fixed modification, while protein N-terminal acetylation, methionine oxidation, and phosphorylation (S, T, Y) were set as variable modifications (MS tolerance of 1.5 Da, an MS/MS tolerance of 0.5 Da, charge states of +1 through +5). Trypsin cleavage was used with up to two missed cleavages allowed, along with a minimum peptide length of 6 amino acids required. The false discovery rate was set to 1% for both peptide and protein identifications. Two technical replicates ($n = 2$) were analyzed. Composite MS/MS coverage for $h$MTHFR was 82%.

## Crystallization of $c$MTHFR

Crystals of the inhibited form (T-state) of $c$MTHFR were grown as follows: Fifty mg/mL (~210 μM) $c$MTHFR$^{R315A}$ was prepared in 50 mM KPB, pH 7.4 containing 500 μM AdoMet and 250 μM FAD. The protein solution was mixed in a 1:1 ratio with the reservoir solution (0.1 M sodium acetate, pH 4.8, 0.2 M ammonium sulfate, 6% polyethylene glycol monomethyl ether 2000). Crystals were grown at 20 °C using the hanging-drop vapor diffusion method. Crystals of $c$MTHFR in the active form (R-state) were obtained using a $c$MTHFR Glu21Gln, Leu393Met, Val516Phe mutant ($c$MTHFR$^{E21Q, L393M, V516F}$), whose surface lysine residues were modified via reductive methylation. Twenty mg/mL of $c$MTHFR$^{E21Q, L393M, V516F}$ was prepared in 25 mM Tris, pH 7.4, containing 50 mM KCl, 500 μM FAD, and 1 mM TCEP. The protein solution was mixed in a 1:1 ratio with the reservoir solution (0.1 M HEPES, pH 7.5, 0.1 mM potassium chloride, 20 mM magnesium chloride, 22% poly(-acrylic acid sodium salt) 5100). Crystals were grown at 20 °C using the sitting-drop vapor diffusion technique. Crystals were briefly transferred to a cryo-protectant solution (the reservoir solution containing 20% glycerol), 1 min for T-state crystals, and 30 sec for R-state crystals, before harvesting and flash freezing in liquid nitrogen.

Data collection and processing statistics are summarized in Supplementary Table 1. Data for $c$MTHFR$^{R315A}$ were indexed to space group $P22_12_1$ (unit-cell parameters $a = 130.66$, $b = 149.95$, $c = 171.06$ Å) with four molecules in the asymmetric unit (Matthew's coefficient VM = 2.99 Å$^3$ Da$^{-1}$, 59% solvent content). Data for $c$MTHFR$^{E21Q, L393M, V516F}$ were indexed to space group $P2_12_12_1$ (unit-cell parameters $a = 117.97$, $b = 151.38$, $c = 188.05$ Å) with four molecules in the asymmetric unit (Matthew's coefficient VM = 3.00 Å$^3$ Da$^{-1}$, 59% solvent content).

## Data collection and refinement

X-ray data sets were collected at 100 K on LS-CAT beamline 21-ID-D at the Advanced Photon Source, Argonne National Laboratory (Argonne, IL). Data sets were processed using xia2/DIALS[40]. Initial phases for $c$MTHFR$^{R315A}$ were obtained using Phaser[41]. The catalytic domain of $h$MTHFR (6FCX) and the regulatory domain of $h$MTHFR (6FCX) without ligands were used as an N-terminal and C-terminal rigid body, respectively, as search models for molecular replacement. Iterative model building and corrections were performed manually using Coot[42] following molecular replacement, with the loop region being placed manually as a poly-alanine chain, and subsequent structure refinement was performed with CCP4 Refmac5[43]. PDB-REDO[44] was used to assess the model quality in between refinements and to fix any rotamer and density fit outliers automatically. The model quality was evaluated using MolProbity[45]. For $c$MTHFR$^{E21Q,L393M,V516F}$, initial phases were obtained via molecular replacement using MRBUMP[46]. The search model used was an AlphaFold model of an MTHFR homolog (UniProt A0A175W7M4). Iterative model building and corrections were performed manually using Coot[42] following molecular replacement, and subsequent structure refinement was performed with CCP4 Refmac5[43]. Initial refinement was conducted using BUSTER[47] to rapidly fix Ramachandran, rotamer, and density fit outliers, refining to convergence and adding water molecules in the final automated round of refinement. Phenix eLBOW[48] was used to generate the initial ligand restraints using ligand ID "FAD". Phenix LigandFit[49] was used to provide initial fits and placements of the FAD ligands. PDB-REDO[44] was used to assess the model quality in between refinements, to fix any rotamer and density fit outliers, and to optimize the ligand geometry and density fit automatically. Figures showing crystal structures were generated in PyMOL[50].

## Statistical analysis and reproducibility

Unless otherwise stated, functional assays were conducted using $n = 2$ independent replicates. At least three independent experiments were conducted for each functional assay. All attempts at replication were successful. Analysis and curve-fitting was performed using Prism 10.2.3.

**Reporting summary**

Further information on research design is available in the Nature Portfolio Reporting Summary linked to this article.

## Data availability

The structure coordinates and structure factors reported in this study have been deposited in the Protein Data Bank under accession codes 8UY1 (cMTHFR$^{E21Q, L393M, V516F}$) and 8UY2 (cMTHFR$^{R315A}$). The PDB code of the previously published structure used in this study is 6FCX. The mass spectrometry phosphorylation data set and raw files have been deposited to the MassIVE repository under accession number MSV000094828 [https://doi.org/10.25345/C5DZ03C7F]. All other data are available from the corresponding authors upon request. Source data are provided with this paper.

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

## Acknowledgements
The authors acknowledge the LS-CAT beamline at the Advanced Photon Source for beamtime. Figure cartoons (Fig. 4b, c, Fig. 8, Supplementary Fig. 6c, and Supplementary Fig. 7c) were created with BioRender.com. KY thanks the program "Improvement of Research Environment for Young Researchers" and Sayaka Igari for her assistance at the Tokyo University of Agriculture and Technology in obtaining the present study's preliminary findings. This work was funded by Rackham Merit Fellowship (J.M.) and the National Science Foundation (CAREER 194517 to M.K.).

## Author contributions
K.Y. wrote the original draft; J.M. and M.K. contributed to writing and editing the paper. K.Y. and J.M. expressed and purified proteins and performed biochemical analysis. K.Y. and J.M. contributed to the protein crystallization. K.Y. and M.K. collected crystallographic data. J.M., K.Y., and M.K. solved the structures, and J.M., K.Y., and M.K. performed structural analysis. All authors edited and approved the final version of the manuscript.

## Competing interests
The authors declare no competing interests.

## Additional information
**Supplementary information** The online version contains show supplementary material available at https://doi.org/10.1038/s41467-024-49327-5.

