## [Peer Review File · Nature Communications]

Structural basis of S-adenosylmethionine-dependent Allosteric Transition from Active to Inactive States in Methylenetetrahydrofolate ReductaseREVIEWER COMMENTS

Reviewer #1 (Remarks to the Author):

Methylenetetrahydrofolate Reductase (MTHFR) stands as one of the most extensively studied enzymes. Despite the wealth of work performed, the allosteric regulation mechanism of the eukaryotic MTHFR remains mostly unknown. In this manuscript, Yamada et al. tackle this issue, revealing the essentials of the enzyme's regulation. Their strategy was well-thought-out: unable to crystallize the human protein, they shifted their focus to eukaryotic homologs suitable as model systems. Consequently, a fungal enzyme was crystallized for the first time, capturing the inactive T-state. The experimental structure beautifully unveils the intricacy of the regulatory mechanism. Two AdoMet molecules bind to the enzyme's regulatory domain, initiating a likely stepwise conformational change that rearranges a domain-linker region. This wedges into the enzyme's active site, creating an occluded and inhibited conformation. The authors excelled in both experiment quality and describing the complex structural changes underlying the R-to-T transition. This paper promises to be a landmark in enzymology, specifically in folate metabolism.

Several points need attention:

Initially, I was somewhat misled by the section on the human enzyme because the introduction emphasizes the fungal protein. Please add a couple of sentences about the project's logic, explaining why it started with the human enzyme and its pathological mutants.

The text and experiments about phosphorylation (lines 102-130) should be mostly removed as phosphorylation is not addressed in this manuscript. Move it to the Supplementary Information (SI). The main text should simply mention the authors' attempt to crystallize the human enzyme and the discovery of new phosphorylation sites.

Line 131 should explicitly state that the Arg357Cys mutation was chosen based on published data.

In Figure 2 and associated text, provide steady-state k_{cat} and K_M parameters for the fungal enzyme. The Figure 2 legend should list the enzyme and substrate concentrations used to obtain the curves and spectra shown.

In Figure 3, the legend should indicate which ligands are bound and the coloring used to depict them.

Lines 200-215: Move the limited proteolysis data text after presenting the 3D structure. Use it as an experiment validating the relevance in solution of the described conformational changes.

Line 230: The biochemical properties of Arg315Ala should be minimally described (activity, spectra).

The methods indicate that R-state (not the T-state?) crystallisation was done with a triple E21Q, L393M, V516F mutant where the lysines were vhemically methylated. Please explain this in the main text.

Lines 261 and following. "The alignment....is indistinguishable.?" Rephrase for clarity.

Figure 5 is confusing due to a likely mistake in residue numbering. For instance, the legend states that Arg357 interacts with Glu360, whereas the figures outline an interaction between Arg358 and Glu360. Thr368 is not shown in the figure.

Related to the above point: The manuscript is sometimes challenging to follow due to the back-and-forth between the residue numbering of the human and fungal enzyme (e.g., lines 290-300). I recommend using the residue numbering of the fungal enzyme for consistency.

Reviewer #2 (Remarks to the Author):

The following paper reports on the structures of R-state and T-state (inhibited) form of MTHFR from the fungus *Chaetomium thermophilum*. cMTHFR serves as a suitable model for understanding the multifaceted allosteric regulation of mammalian MTHFR, and important enzyme that links one carbon folate and methionine metabolism. The work is significant because (i) it provides a structural rationale for how 2 molecules of SAM act as allosteric inhibitors and (ii) reveals a new mechanism for allosteric control of an enzyme, namely the interweaving of mobile linker region between the regulatory and catalytic domains of the enzyme. The structures also lead to a proposal for how phosphorylated residues at the N-terminus contributes to stabilization of the T-state. The work also supports or explains published biochemical/biophysical data for MTHFR, for example perturbation of the FAD absorbance spectra induced by SAM.

The poor atomic resolution of both the R- and T-state of the cMTHFR structures is a limitation of the study. The resolution is suitable for tracking the reorganization of domain, but perhaps not for mapping intermolecular interactions. Can you show an H-bond between R358 and Glu360 at 3.4 Å resolution?

Biochemical data supporting the binding of two equivalent of SAM would confirm that the second binding site is physiologically relevant or at least happens in solution. Can this be done with ITC? Given that site 2 is occluded in the R state and site 1 is not, they would presumably have different binding affinities. You could measure SAM binding to wild type cMTHFR and the R315 mutant.

Line 63, Fig 1B What is the % of sequence identity between the cMTHFR and human MTHFR. Does the 38% just compare the catalytic domain and regulatory domains, without the linker?

Line 154. Reference 21 did not appear measure hMTHFR catalytic preference for NADPH versus NADH.

Fig 2a. Why was free FAD added? Why did it slow the initial velocity of the reaction? Provide more information in the figure legend for 2a and in Results section. Catalytic preference for NADPH vs NADH should be a ratio of their k_{cat}/K_m values.

Fig 2b. Explain the colouring of the initial velocity traces shown in Fig 2b (is one with SAM?) Does SAM have an inhibitory effect on R315C? If not, this would support its T-state conformation.

Did you subject the R315C variant to limited proteolysis (with and without SAM). This would support its stabilization of the T-state.

Line 227; Abolishment of activity (?) or a significantly reduced activity. The R315C still shows activity.

It seems odd to report on the biochemical properties of R315C but determine the structure of R315A.

Figure 5. The labelling of the residues appears off. Is Arg358 supposed to be Arg357 and is Arg357 supposed to Arg 377 in the zoomed in image?

Figure 5. Why is Tyr403 shown in Fig 5? It is not referred to in the Figure legend or text.

Line 283. "occludes access to FAD via Tyr361" There isn't a Tyr361 in the sequence.

Figure 6. Is it Tyr361 or Tyr403? (how conserved is this residue?)

Line 312 and elsewhere, is it Tyr361 or Tyr403?

Line 319. "abolished catalytic activity and retention of the FAD cofactor observed with cMTHFR315A". This was not experimentally shown.

Line 382: "phosphorylation status and effector-induced and stabilized conformational changes," remove the second "and"

Supporting information

Line 92. Complete the following sentence: Patients with 1081C>T have 92 low MTHFR activities, 5~27% of controls; thus, most of them are in severe (usually the activity range is 0%-20% of controls MTHFR deficiency^{9,10} 93 .

Line 99: Please specify the mutant in the following sentence: "Therefore, it is likely that the mutant

undergoes a post-translational modification, such as phosphorylation

Structural basis of S-adenosylmethionine-dependent Allosteric Transition from Active to Inactive States in Methylenetetrahydrofolate Reductase

Manuscript Number: NCOMMS-23-59767

Response to reviewers

(Referee comments in black; responses in red)

Referee 1

1. Initially, I was somewhat misled by the section on the human enzyme because the introduction emphasizes the fungal protein. Please add a couple of sentences about the project's logic, explaining why it started with the human enzyme and its pathological mutants.

We appreciate the reviewer's comments and therefore have included a couple of sentences in the introduction to state that the overarching goal was to understand the allosteric regulation of MTHFR biochemically and structurally. Initially, the human protein was studied and while extremely valuable biochemical data was collected, it proved recalcitrant to structural studies, leading us to pivot to a eukaryotic MTHFR model in the fungal protein. The introduction has been edited to include the specific rationale for starting with the human MTHFR and why the fungal protein MTHFR homolog/model system was ultimately focused on.

2. The text and experiments about phosphorylation (lines 102-130) should be mostly removed as phosphorylation is not addressed in this manuscript. Move it to the Supplementary Information (SI). The main text should simply mention the authors' attempt to crystallize the human enzyme and the discovery of new phosphorylation sites.

We agree with the reviewer's comment and have moved this section (lines 102-107) to the Supplementary Information (SI Figs. 1 and 2. Identification of phosphorylation sites of recombinant wild-type *h*MTHFR) to emphasize the discovery of the three new phosphorylation sites. We have condensed the following section (lines 123-130) to highlight the importance of the biochemical data obtained from the human MTHFR and added references to rationalize the use and study of specific patient mutations. Additionally, a plot containing summarized biochemical data for patient mutations has been added to the Supplementary Information (SI Fig. 4).

3. Line 131 should explicitly state that the Arg357Cys mutation was chosen based on published data.

We agree with the reviewer's comment and have added the relevant reference in the main text, along with a brief sentence summarizing the specific mutation: "The Arg357Cys patient mutant is based on rare mutation (1081C>T, Arg357Cys)" (Goyette in *Am. J. Hum. Genet.* **59**, 1268-1275 (1996)).

4. In Figure 2 and associated text, provide steady-state k_{cat} and K_M parameters for the fungal enzyme. The Figure 2 legend should list the enzyme and substrate concentrations used to obtain the curves and spectra shown.

Specific values for the concentrations of *c*MTHFR used and relevant substrate concentrations have been added to the figure legend (50 nM protein for panels a, b, and d) (100 μ M of NADPH or NADH for panel a; for additions in panel a, 100 μ M NADPH combined with 2 μ M FAD or 100 μ M AdoMet) (10 μ M protein for panel c along with 0-200 μ M AdoMet for panel c). The methods have been lightly edited to fix a transcription error regarding the amount of protein used to obtain the data in panel c (fixed to 10 μ M).

With regards to providing steady-state kinetic parameters, the assay used (NADPH:menadione oxidoreductase assay) is *not under steady-state* conditions. Additionally, and more importantly, MTHFR is an allosteric enzyme, and thus does not obey standard Michaelis-Menten kinetics. While we are actively engaged in a more detailed kinetic analysis of this enzyme, we believe its discussion falls outside the scope of this paper. The main goal behind Figure 2 was to establish the fungal protein's suitability as a biochemical model for eukaryotic MTHFRs. Our preliminary data suggest that the mechanism of AdoMet inhibition, particularly the binding kinetics in solution for *c*MTHFR, will align with those previously conducted by Jencks and Matthews (*JBC* 1986).

5. In Figure 3, the legend should indicate which ligands are bound and the coloring used to depict them.

An explicit mention of the ligands bound to each structure and their respective colors have been included, and the Figure 3 legend has been revised to explicitly mention that the R-state *c*MTHFR structure does not contain AdoHcy bound.

6. Lines 200-215: Move the limited proteolysis data text after presenting the 3D structure. Use it as an experiment validating the relevance in solution of the described conformational changes.

We agree with the reviewer's suggestion and have moved this section after Figure 3 is introduced and edited the text to emphasize how the limited proteolysis data provides a biochemical validation of the conformational dynamics in solution.

7. Line 230: The biochemical properties of Arg315Ala should be minimally described (activity, spectra).

Spectra for the R315A mutant have been added to Fig. 2d. There is no discernable difference between the R315C and R315A mutants, spectroscopically or biochemically with regards to their activity (R315C: 14% relative to wild-type, R315A: ~14.7% relative to wild-type). Notably, the R315A mutant displays the same FAD absorbance quench and red-shift at 450 nm, altering the absorbance maxima of FAD from 453 nm to 463 nm.

8. The methods indicate that R-state (not the T-state?) crystallisation was done with a triple E21Q, L393M, V516F mutant where the lysines were chemically methylated. Please explain this in the main text.

We have added a couple of sentences in the main text stating the use of the triple mutant: L393M and V516F were introduced to disrupt AdoMet binding, as previous work with the human enzyme has shown that AdoMet was copurified during protein purification (*Froese et al. (2018)*), which could explain why no AdoHcy was found bound in our R-state structure; E21Q mutant was introduced in an attempt to co-crystallize cMTHFR in the R-state with CH₃H₄-folate, but we were not successful. Additionally, a rationale is also provided behind the use of reductive lysine methylation to aid in crystallization, along with relevant references regarding the use of this method to enhance the crystallizability of proteins.

Minor textual suggestions:

9. Lines 261 and following. "The alignment....is indistinguishable.."? Rephrase for clarity.

The relevant text has been edited to highlight that the structures of individual MTHFR domains as compared via superposition of (regulatory vs. regulatory, catalytic vs. catalytic) between the R and T-states demonstrates that they are structurally rigid and maintain the same topology; in other words, the regulatory domain of the R and T-state superimpose with an RMSD of 0.53 Å and the catalytic domains with an RMSD of 0.39 Å. This indicates that flexibility and conformational

dynamics are not mediated by MTHFR domain(s) restructuring, but by rearrangement of the linker connecting the domains.

10. Figure 5 is confusing due to a likely mistake in residue numbering. For instance, the legend states that Arg357 interacts with Glu360, whereas the figures outline an interaction between Arg358 and Glu360. Thr368 is not shown in the figure.

We appreciate the reviewer pointing this out and have fixed the residue numbering mistakes and included the outlined interactions in Figure 5. The figure legend has been changed to reflect the residue numbering of the fungal enzyme.

11. Related to the above point: The manuscript is sometimes challenging to follow due to the back-and-forth between the residue numbering of the human and fungal enzyme (e.g., lines 290-300). I recommend using the residue numbering of the fungal enzyme for consistency.

We appreciate the reviewer pointing out this source of confusion and have revised the main text to use the residue numbering of the fungal enzyme for consistency. The residue numbering corresponding to the human enzymes has been left solely for the results section regarding the biochemical characterization of human MTHFR and for Figure 8 and the model presented for the human MTHFR allosteric regulation (this could be changed).

Referee 2

1. The poor atomic resolution of both the R- and T-state of the cMTHFR structures is a limitation of the study. The resolution is suitable for tracking the reorganization of domain, but perhaps not for mapping intermolecular interactions. Can you show an H-bond between R358 and Glu360 at 3.4 Å resolution?

We value and acknowledge the reviewer's comments regarding the limitations of mapping intermolecular H-bonding interactions given the resolution of our R-state structure. While the figure in question (Figure 5) shows the human R-state structure (6FCX) which has a higher-resolution (2.50 Å), we have changed the figure to reflect our R-state structure and residue numbering for clarity and consistency.

In addition, we have edited the text and figure to emphasize our focus on the distribution of charge of the residues, particularly those surrounding the linker and R315. This focus is crucial as the

R315A mutant has proven sufficient for capturing and elucidating the structure of the elusive T-state. A comparison between the human R-State structure and our R-state structure has been included in the Supplementary Information (SI Fig. 12).

2. Biochemical data supporting the binding of two equivalent of SAM would confirm that the second binding site is physiologically relevant or at least happens in solution. Can this be done with ITC? Given that site 2 is occluded in the R state and site 1 is not, they would presumably have different binding affinities. You could measure SAM binding to wild type cMTHFR and the R315 mutant.

We greatly value the reviewer's comments highlighting the unusual stoichiometry observed for AdoMet/SAM binding. For over a year, we have dedicated extensive efforts to determine binding affinities for AdoMet using both the wild-type enzyme and the R315 mutant, employing native (gas-phase) mass spectrometry (IM-MS) and conducted equilibrium binding measurements using ITC, as recommended by reviewer 2. However, these efforts have not yet provided conclusive data. However, despite these efforts, conclusive data remains elusive at this time Using IM-MS, the wildtype and R315A mutant were found to be highly heterogenous, and individual contributions between states in the presence and absence of ligand could not be deconvoluted. In our hands, the best sample and runs confirmed that the wildtype is found as a dimer, with two FADs bound.

Froese *et al.* 2018 demonstrated using native MS that as-purified enzyme has AdoMet bound: their N-terminal truncated construct, which lacks the phosphorylation site/region showed up to two AdoHcy bound, but they attributed this to AdoMet degradation; full-length hMTHFR showed up to 2 AdoMet molecules bound, but upon phosphatase treatment, they saw 2 AdoHcy molecules. They argue that this was the result of AdoMet degradation to AdoHcy. Given that cMTHFR lacks the N-terminal phosphorylation region, it is possible that AdoMet could likewise be degraded.

During the submission of our work, we came across a preprint (Froese *et al.*, bioRxiv, 2024) of a study published this month in *Nat. Comm.* by the same group that investigated human MTHFR and its allosteric regulation, leading to the discovery of the first human MTHFR T-state structure. They obtained results that complement and validate ours, finding that two molecules of AdoMet are bound in their T-state structure. In their new study, the authors were able to conduct ITC

experiments with the human enzyme and determined that the binding stoichiometry of AdoMet was double that of AdoHcy, indicating that the 2:1 stoichiometry observed in the T-state structure is not an artifact. However, while they were able to provide K_d values, the binding affinities of each site could not be determined separately. Additionally, the question of cooperativity was not addressed.

Jencks and Mathews (JBC, 1986) determined/observed the spectrophotometric titration of AdoMet to MTHFR and tracked the subsequent absorbance quench of FAD as a function of AdoMet concentration, finding that the resulting curve was sigmoidal/biphasic in nature and posited: “The two phases could be due to either fast AdoMet binding to the R state followed by a slower isomerization to, e.g. AdoMet-ligated T state, or binding of first one and then a second AdoMet to R state enzyme. Our model adopts the first of these explanations.”

Our initial AdoMet titration to *c*MTHFR wild-type in Fig. 2c was fit using a Michaelis-Menten/hyperbolic curve; however, fitting the same data using a sigmoidal curve/Hill plot gave a superior fit ($R^2 = 0.9831$ vs. $R^2 = 0.9568$), with Hill coefficient equal to 2.09, and an IC_{50} of 18.78 μ M; the sigmoidal nature of the FAD quench and the positive Hill coefficient suggests that AdoMet binding is cooperative, and that there are thus multiple AdoMet binding events, the first of which facilitates the binding of the second AdoMet molecule. The “slow” step observed and first reported by Jencks and Mathews is thus best explained by the second of their two hypotheses: that there are two AdoMet binding events, and that the binding of the first AdoMet molecule triggers the R-T state transition by cooperatively enabling the binding of a second AdoMet molecule. The observed positive cooperativity validates our structural analysis, since binding of the first AdoMet is necessary to reveal the cryptic secondary AdoMet site.

To determine AdoMet binding stoichiometry, Jencks and Matthews (1986) performed spectrophotometric titration of AdoMet and tracked $Abs_{450-503}$ vs. [AdoMet]. They showed that their data is best fit by a theoretical model “computed for a model with two AdoMet-binding sites, where AdoMet binding to R or T state subunits produces independent effects on the spectrum.” Our data mimics this model:

Figure 1. Spectrophotometric titration of cMTHFRwt with AdoMet. UV-Vis spectral changes of cMTHFR^{wt} (10 μM) titrated with varying concentrations of AdoMet (0-200 μM). Absorbance changes were monitored at 503 and 450 nm, with complete spectra also recorded. The absorbance changes were similar to those reported previously (Jencks and Matthews, 1986): “a decrease in the flavin absorbance at 450 nm, an increase in the long wavelength shoulder of the 450 nm band (maximal at 503 nm) and an isosbestic point at 485 nm”. The best-fit curve is a 4-parameter logistic (4PL) model, with a Hill coefficient of 2.318 and IC₅₀ of 19.72 (>99% confidence, Akaike’s information criterion implemented in Prism, R²: 0.9934). 95% confidence intervals are shown as dashed lines.

The only other enzyme that binds two molecules of AdoMet for allostery is, to our knowledge, *Arabidopsis Thaliana* threonine synthase, where AdoMet acts as an allosteric activator. Using a similar approach, Curien *et al.* (*Biochemistry*, 1998) showed spectrophotometrically that AdoMet binding was cooperative, with a Hill coefficient of 2: “Since such a value represents the maximal value that can be obtained for a dimeric enzyme both subunits are strongly constrained by their association, suggesting that they change their conformation upon SAM binding in a concerted manner.”

As such, while we have been unable to obtain binding affinities for AdoMet to either the wild-type or the R315 mutant, we believe that the spectrophotometric data indicate that the observation of two molecules of AdoMet in our T-state structure is not an artifact, and that we do indeed observe multiple AdoMet binding events in solution. Froese *et al.*’s results confirm our findings, and

together, they complement each other. Although we are currently working on discerning the specific binding affinities of each AdoMet event, we feel that further delving into this aspect is outside the purview of this paper.

3. Line 63, Fig 1B What is the % of sequence identity between the cMTHFR and human MTHFR. Does the 38% just compare the catalytic domain and regulatory domains, without the linker?

We appreciate the reviewer pointing out this source of confusion. The values only refer to the domains, not the linker. The figure has been revised for clarity.

4. Line 154. Reference 21 did not appear measure hMTHFR catalytic preference for NADPH versus NADH.

We appreciate the reviewer for catching this mistake. Reference 21 was removed and in its place, Reference 7 was used. In addition, the sentence has been edited to include References 1 and 22 regarding the NADPH preference observed in rat and pig MTHFR, respectively.

5. Fig 2a. Why was free FAD added? Why did it slow the initial velocity of the reaction? Provide more information in the figure legend for 2a and in Results section. Catalytic preference for NADPH vs NADH should be a ratio of their k_{cat}/K_M values.

Previous work on human MTHFR had indicated that FAD is readily released in dilute solution. As such, excess FAD was added to ascertain its effect on activity. Figure 2a shows that the addition of excess FAD (2 μM) to the reaction mixture *accelerates* the consumption of NADPH, at $\sim 150 \mu\text{M}/\text{min}$ (versus $120 \mu\text{M}/\text{min}$). The figure legend and results section have been edited to reflect these values and for clarity.

With regards to catalytic preference and the aforementioned units (k_{cat}/K_M), we would like to note that the assay used (NADPH:menadione oxidoreductase assay) is not under steady-state conditions. Additionally, and more importantly, MTHFR is an allosteric enzyme, and thus does not obey standard Michaelis-Menten kinetics. As such, while a more detailed kinetic analysis of this enzyme is currently underway, we believe it is beyond the scope of this paper. The main goal behind Figure 2 was to demonstrate the validity of the fungal protein as a biochemical model for eukaryotic MTHFRs.

6. Fig 2b. Explain the colouring of the initial velocity traces shown in Fig 2b (is one with SAM?) Does SAM have an inhibitory effect on R315C? If not, this would support its T-state conformation.

We are grateful for the reviewer highlighting this source of confusion. There is no SAM/AdoMet added in the traces for Fig. 2b. The figure has been edited to include a legend for the coloring and the figure legend likewise edited to clarify that there is no AdoMet added.

The R315C mutant showed 14% activity relative to the wild-type (line 160). The effect of AdoMet on the activity of R315C/A was explored but the observed activity was too low for our assay conditions; tentatively, addition of any amount of AdoMet (above 1 μ M) lead to minimal activity that could be a result of NADPH rescue, as it is an allosteric regulator that can reverse AdoMet inhibition. However, the activity was so low as to be virtually indistinguishable from background.

7. Did you subject the R315C variant to limited proteolysis (with and without SAM). This would support its stabilization of the T-state.

While the R315C mutant was not subjected to limited proteolysis with and without SAM, the R315A mutant was. The resulting gel has been added to the Supplementary Information (SI Fig. 7) and shows indistinguishable digest patterns that are AdoMet-*independent*. A few sentences have been added in the main text to highlight how this supports that the R315A mutant is stabilized in the T-state. We appreciate the reviewer for bringing up this point and for their insight.

8. Line 227; Abolishment of activity (?) or a significantly reduced activity. The R315C still shows activity.

While we agree with the reviewer that the R315C mutant still shows residual activity, we believe that this is due to the allosteric activation caused by NADPH. In the NADPH-menadione assay, AdoMet inhibition can be partially rescued by NADPH, which acts in a diametrically opposed manner. As such, the presence of NADPH serves to shift the R/T equilibrium to the R-state. However, the R315C is preferentially locked in the T-state, so the rescue effect of NADPH addition is minimal.

9. It seems odd to report on the biochemical properties of R315C but determine the structure of R315A.

The R315C mutant was not used for crystallization to avoid the potential introduction of disulfide bonds. Given that R315 lies on the linker region and is presumably solvent exposed in the T-state, we rationalized that using the R315A mutant would avoid the potentially unfavorable reactivity

associated with introducing a thiol to a solvent-exposed region, while increasing the likelihood that the protein could be crystallized.

10. Figure 5. The labelling of the residues appears off. Is Arg358 supposed to be Arg357 and is Arg357 supposed to be Arg 377 in the zoomed in image?

We are thankful to the reviewer for identifying this error. Figure 5 has been edited to fix the residue numbering and in line with the other reviewer's comments the fungal enzyme residue numbering has been used for the sake of clarity and consistency.

11. Figure 5. Why is Tyr403 shown in Fig 5? It is not referred to in the Figure legend or text.

We appreciate the reviewer's comment and advice. Figure 5 has been edited to remove the mentioned Tyr residue.

12. Line 283. "occludes access to FAD via Tyr361" There isn't a Tyr361 in the sequence.

We appreciate the reviewer pointing out this source of confusion. The fungal enzyme does indeed have a Tyr361 residue in its sequence, while the human enzyme does not. For clarity and consistency, the residue numbering throughout the text has been revised to use the fungal enzyme, unless otherwise noted.

13. Figure 6. Is it Tyr361 or Tyr403? (how conserved is this residue?)

As shown in Figure 5c, the Tyr residue is not conserved in *Saccharomyces cerevisiae* Met13. The residue is highly conserved: a corresponding Weblogo representation of a multiple sequence alignment generated using DeepMSA2 (Zheng *et al. Nat. Methods* (2024)) has been added to the Supplementary Information for reference (SI Fig. 13).

14. Line 312 and elsewhere, is it Tyr361 or Tyr403?

It is Tyr361 in the fungal MTHFR, and Tyr403 in the human MTHFR. We have edited the text to focus on the residue numbering corresponding to the fungal MTHFR for clarity and consistency.

15. Line 319. "abolished catalytic activity and retention of the FAD cofactor observed with cMTHFR315A". This was not experimentally shown.

The sentence has been edited to reflect that this was demonstrated with the human MTHFR R357C mutant and that the abolished activity was confirmed for the cMTHFR R315C mutant, with the presumption is that the abolished activity and retention of the FAD cofactor would hold true for

the cMTHFR R315A mutant as well (preliminarily, the activity relative to the wild-type does, R315A: ~14.7%, R315C: 14%).

Minor textual suggestions:

17. Line 382: “phosphorylation status and effector-induced and stabilized conformational changes,”, remove the second “and”

We appreciate the reviewer for catching this mistake. The sentence has been edited as requested.

Supplementary Information:

18. Line 92. Complete the following sentence: Patients with 1081C>T have 92 low MTHFR activities, 5~27% of controls; thus, most of them are in severe (usually the activity range is 0%-20% of controls MTHFR deficiency^{9,10} 93 .

We appreciate the reviewer for catching this mistake. The sentence has been edited accordingly.

19. Line 99: Please specify the mutant in the following sentence: “Therefore, it is likely that the mutant undergoes a post-translational modification, such as phosphorylation

We appreciate the reviewer for catching this. The mutant in question is Arg357Cys, and the text has been edited as requested.

REVIEWERS' COMMENTS

Reviewer #1 (Remarks to the Author):

The manuscript was carefully revised and improved, following the reviewers' suggestions and comments. The answer to my point about the steady-state kinetics is convincing. This work will be a milestone in the field.

Minor comment

I found the sentence "Thus, loss of Arg315 in the patient mutant can lead to destabilization of the retractable-hinge region and in turn of the T-state" (right before Figure 6) misleading. The loss of Arg315 should stabilise the T-state.

Reviewer #2 (Remarks to the Author):

The manuscript is more clear and suitable for publication.